Identification of significant genes signatures and prognostic biomarkers in cervical squamous carcinoma via bioinformatic data

He Yunan 1
Hu Shunjie 2
Zhong Jiaojiao 3
Cheng Anran 4
Shan Nianchun shannianchun@163.com 5
1 Department of Gynecology and Obstetrics, Sun Yat-sen Memorial Hospital, Sun Yat-sen University , Guangzhou , Guangdong , China
2 Zhongshan School of Medicine, Sun Yat-sen University , Guangzhou , Guangdong , China
3 Department of Dermatology, Sun Yat-sen Memorial Hospital, Sun Yat-sen University , Guangzhou , Guangdong , China
4 Department of Gynecology Oncology, Sun Yat-sen University Cancer Center, State Key Laboratory of Oncology in South China, Collaborative Innovation Center for Cancer Medicine , Guangzhou , Guangdong , China
5 Departmen of Gynecology and Obstetrics, Xiangya Hospital, Central South University , Changsha , Hunan , China
Ge Jianye
Electronic publication date: 2020 Dec 2
Publication date: 2020
Volume: 8
Electronic Location ID: e10386
Received 2020 Jul 28; Accepted 2020 Oct 27
Copyright: ©2020 He et al.
Copyright year: 2020
Copyright holder: He et al.
License: This is an open access article distributed under the terms of the Creative Commons Attribution License, which permits unrestricted use, distribution, reproduction and adaptation in any medium and for any purpose provided that it is properly attributed. For attribution, the original author(s), title, publication source (PeerJ) and either DOI or URL of the article must be cited.
License URL: https://creativecommons.org/licenses/by/4.0/

Keywords: Cervical squamous carcinoma, Bioinformatics, RFC4, Prognostic biomarker

Funding: The authors received no funding for this work.

==============================
Background

Cervical squamous cancer (CESC) is an intractable gynecological malignancy because of its high mortality rate and difficulty in early diagnosis. Several biomarkers have been found to predict the prognose of CESC using bioinformatics methods, but they still lack clinical effectiveness. Most of the existing bioinformatic studies only focus on the changes of oncogenes but neglect the differences on the protein level and molecular biology validation are rarely conducted.

Methods

Gene set data from the NCBI-GEO database were used in this study to compare the differences of gene and protein levels between normal and cancer tissues through significant pathway selection and core gene signature analysis to screen potential clinical biomarkers of CESC. Subsequently, the molecular and protein levels of clinical samples were verified by quantitative transcription PCR, western blot and immunohistochemistry.

Results

Three differentially expressed genes (RFC4, MCM2, TOP2A) were found to have a significant survival (P < 0.05) and highly expressed in CESC tissues. Molecular biological verification using quantitative reverse transcribed PCR, western blotting and immunohistochemistry assays exhibited significant differences in the expression of RFC4 between CESC and para-cancerous tissues (P < 0.05).

Conclusion

This study identified three potential biomarkers (RFC4, MCM2, TOP2A) of CESC which may be useful to clarify the underlying mechanisms of CESC and predict the prognosis of CESC patients.

Introduction

Cervical cancer now ranks fourth in the most prevalent cancers and it is the most common gynecological cancer in developing countries (Vu et al., 2018). Despite the increase in the incidence of cervical adenocarcinoma, cervical squamous carcinoma (CESC) is still the most common type of cervical cancer (Wang et al., 2004; Galic et al., 2012). Currently, a large number of gene mutations have been proved to be related to the pathogenesis of cervical cancer, which can be used as biomarkers for early detection, like DNA mutations occurring on the oncogenes tumor protein 53 (TP53) (Crook et al., 1992), phosphatase and tensin homolog (PTEN) (Yang et al., 2015). However, due to the difficulties of early detection and diagnosis, the survival rate of CESC patients still stays weak. Studies also showed that some biological markers can explain the pathogenesis of CESC and predict the consequences of this disease (Mao et al., 2019). Therefore, more reliable biological markers should be explored to comprehensively understanding the pathogenesis of CESC and guide treatment and prognosis.

With the developed bioinformatics and statistical analyses, the potential marker genes can be detected effectively, which shows great strength in the field of discovery and prediction of tumor markers, and plays a guiding role in the treatment and prognosis of the disease (Banwait & Bastola, 2015). Some biomarkers have been found in the field of cervical cancer, such as MicoRNA-425-5p and MicoRNA-489, which have been proposed for prognostic prediction (Sun et al., 2017; Juan et al., 2018).

However, the presented biomarkers for clinical application are far from enough, and in the previous bioinformatics studies, most studies only focus on the changes of oncogenes, which increases the possibility of clinical inefficacy. On the basis of learning the expression of differential genes between cancer tissues and normal tissues, this study analyzed and compared the difference in protein level between cancer tissue and normal tissue, which provides stronger evidence for the validity of biomarkers found in our bioinformatic research.

Materials & Methods

Information of the microarray data

NCBI-GEO (Gene Expression Omnibus) is known as a free public database of microarray cohort. The gene profiles of GSE27678, GSE39001 and GSE7803 were obtained in this study. The three datasets were on the account of GPL570 platform, GPL201 platform and GPL96 platform, including 14 normal cervical tissues and 28 CESC tissues, 12 normal cervical tissues and 43 CESC tissues, 10 normal cervical tissues and 21 CESC tissues, respectively.

Identification of differentially expressed genes

The differentially expressed genes (DEGs) were analyzed by GEO2R to obtain the number of up-down-regulated genes (Barrett et al., 2013). The genes with —log Fold Change— ≥2 and P <  0. 05 were screened as differentially expressed genes.

Gene Ontology and Kyoto Encyclopedia of Genes and Genomes pathway analyses

Gene Ontology (GO) is an international standardized classification system of gene function, which provides a dynamic updating database to describe the attributes of genes and gene products in organisms (Ashburner et al., 2000). The main biological functions of differentially expressed genes could be determined by GO functional significance enrichment analysis. The GO items with q <  0. 05 were considered to be significantly enriched in DEGs.

The Kyoto Encyclopedia of Genes and Genomes (KEGG) database is a bioinformatics resource for linking genomes to life and the environment (Kanehisa et al., 2017). Based on the KEGG database, the enriched pathway analysis of DEGs was carried out to find out the important pathway.

PPI & module analysis

Cytoscape 3.8.0 is a software that was used for visualization and analyzation of complex network (Shannon et al., 2003). Search Tool for the Retrieval of Interacting Genes/Proteins (STRING) is an application that could conduct protein interaction group research, genome research and proteome research (Doncheva et al., 2019). By mapping the information of DEGs to the STRING, we evaluated the protein-protein interaction (PPI) information of DEGs. Interactions experimentally validated with combined score >0.4 and were selected. Subsequently, we used another tool embedded in the Cytoscape named Molecular Complex Detection (MCODE) to clustering constructed functional module of PPI network (Bader & Hogue, 2003). The MCODE scores were set to be greater than 10 and nodes number more than 6. Functional and pathway enrichment for DEGs in the modules were also conducted, P < 0.05 was considered to have significant difference.

Survival analysis of significant genes in CESC and RNA expression of core genes

Kaplan–Meier (K-M) is a widely used method for estimating the survival rate of cancer patients and “Survival” package was applied in the R studio software (Rich et al., 2010). As for the compare of the magnitude of the difference in survival between the 2 groups, a Cox univariate hazard ratio (HR) was calculated. The clinical significance of each genes was also evaluated by performing the survival analysis of single gene in survival-related gene sets. A log-rank test was used to calculate the statistical significance of the survival difference between these 2 groups mentioned above, and the P value set as 0.05 was considered to be significant.

Gene Expression Profiling Interactive Analysis (GEPIA) is visualization tool for gene research (Tang et al., 2017). In this study, GEPIA was applied to analyze RNA expression of selected genes on the basis of thousands of simples from the TCGA database.

Specimen collection

The tissues or cells of CESC patients were collected from Xiangya Hospital of Central South University in order to verify the high expression of RFC4 in tumor tissues for molecular and protein levels. This study was proved by Medical Ethics Committee of Xiangya Hospital (No. 201912542). CESC Patients and the kin have signed a consent form, agreeing to use cervical tissue for scientific research.

Molecular biological verification of differences in gene expression

CESC tissues and para-cancerous tissues (para-CT) were selected from CESC patients to conduct the molecular validation of RFC4. The expression levels of RFC in CESC patients with different pathological stages were also compared. The pathological stage of I and II are regarded as early stage which including 4 I B1 patients, 7 I B2 patients, 3 I B3 patients, 3 II A1 patients and 1 II A2 patient. Stage III are divided into advanced stage and 17 patients in III C1 stage were included. Total RNA was extracted from CESC tissues and para-CT using Trizol Reagent (RNAiso Plus, TaKaRa, 9109) according to the manufacturer’s protocols, and reverse transcribed into cDNA using a PrimeScript™ RT reagent Kit with gDNA Eraser (TaKaRa, RR047A-1). Gene expression levels were assessed by quantitative reverse transcribed PCR (qRT-PCR) with TB Green™ Premix (Tli RNaseH Plus, TaKaRa, RR820A) and specific primers:

RFC4 forward: 5′-GGCAGCTTTAAGACGTACCATGG-3′;

RFC4 reverse: 5′-TCTGACAGAGGCTTGAAGCGGA-3′.

The β-actin expression was used as the normalization control. Relative mRNA levels are analyzed using 2−ΔΔCt method.

Verification of differences in protein expression

We adopted the cancerous tissues and para-CT of CESC patients to analyze the differences in protein expression by Western Blotting (WB) technology. The samples for WB analysis was separated using SDS-PAGE and transferred onto a PVDF membrane (Roche) which was blocked with 5% nonfat milk in Tris-buffered saline and incubated overnight at 4 °C with target antibodies against the following proteins: Anti-RFC4 antibody (ab156780, Abcam) and Anti-β-Actin antibody (ab115777, Abcam). After three times washing with PBST (10 min for each time), the membrane was incubated with species-appropriate HRP-conjugated secondary antibodies, the fluorescent signals were detected using SageCapture™ imaging system (SAGECREATION company).

Immunohistochemistry (IHC) assays were also performed to detected protein levels in CESC tissues and para-CT. The tissues were performed into 5- µm-thick tissue sections with formalin fixed and paraffin embedded. Subsequently, there sections were deparaffinized and rehydrated with xylene and graded ethanol respectively, followed by heated in antigen retrieval solution (EDTA, PH 9.0) and endogenous peroxidase inactivation with 3% H2O2. After blocking, the samples were incubated overnight at 4 °C with anti-RFC4 antibody (1:100, ab156780, Abcam). Then the slides were treated with the HRP-conjugated secondary antibody and stained with 3, 3′-diaminobenzidine until brown granules appeared in the membrane, cytoplasm, or nucleus. Finally, the sections were counterstained with hematoxylin at room temperature.

Results

Screening for DEGs

Ninety-two cancer tissues and 36 normal tissues were selected from the three datasets in total, with the help of GEO2R tools, 211, 134 and 260 DEGs were extracted from GSE39001, GSE7803 and GSE27678. And Venn diagram was made by the Venn diagram software to investigate the commonly DEGs in all the three datasets. The results showed that there were 25 commonly DEGs in total and 18 of them were down-regulated while 7 were up-regulated (Fig. 1 and Table 1).

Figure 1 Identification of 25 common DEGs in the three datasets (GSE39001, GSE7803 and GSE27678) through Venn diagrams software.

Different color meant different datasets. (A) Seven DEGs were up-regulated in the three datasets (logFC > 2). (B). Eighteen DEGs were down-regulated in three datasets (logFC > −2).

Table 1 25 common DEGs identified from the three datasets.

Expression	Genes Name	
Up-regulated	M2M DTL CDKN2A TOP2A NUSAP1 RFC4 PLOD2	
Down-regulated	EMP1 IGF1 ALOX12 EDN3 PTGDS KRT1 FOSB GREB1 ESR1 PAMR1 CXCL12 HPGD AR MAL CRNN CRISP3 CFD NDN	

Figure 2 GO and KEGG results shows significant signaling pathways of DEGs.

(A) The results of GO analysis for pathways associated with molecular function (MF). (B) The results of GO analysis for pathways associated with cellular component (CC). (C) The results of GO analysis for pathways associated with biological process (BP). (D) The results of KEGG analysis.

Table 2 GO analysis of different expressed genes in CESC.

Expression	Category	Term	Count	%	p-Value	FDR	
Up-regulated	GOTERM_MF_DIRECT	GO:0005524∼ATP binding	10	21.03	2.84E−4	0.270953	
	GOTERM_MF_DIRECT	GO:0003678∼DNA helicase activity	2	4.21	0.012333	11.169161	
	GOTERM_MF_DIRECT	GO:0003688∼DNA replication origin binding	2	4.21	0.018445	16.278756	
	GOTERM_MF_DIRECT	GO:0003682∼chromatin binding	4	8.41	0.021525	18.752384	
	GOTERM_CC_DIRECT	GO:0030496∼midbody	6	12.62	2.68E−7	2.65E−4	
	GOTERM_CC_DIRECT	GO:0005737∼cytoplasm	14	29.44	1.50E−4	0.147574	
	GOTERM_CC_DIRECT	GO:0005876∼spindle microtubule	3	6.31	0.001486	1.457975	
	GOTERM_CC_DIRECT	GO:0005654∼nucleoplasm	8	16.82	0.005554	5.351061	
	GOTERM_CC_DIRECT	GO:0000784∼nuclear chromosome, telomeric region	3	6.31	0.063091	9.946422	
	GOTERM_CC_DIRECT	GO:0072687∼meiotic spindle	2	4.21	0.014282	13.241928	
	GOTERM_CC_DIRECT	GO:0042555∼MCM complex	2	4.21	0.014282	13.241928	
	GOTERM_CC_DIRECT	GO:0005680∼anaphase-promoting complex	2	4.21	0.035339	29.901598	
	GOTERM_CC_DIRECT	GO:0005634∼nucleus	9	18.93	0.049470	39.406713	
	GOTERM_CC_DIRECT	GO:0072686∼mitotic spindle	2	4.21	0.054262	42.356169	
	GOTERM_CC_DIRECT	GO:0005819∼spindle	2	4.21	0.062745	47.263108	
	GOTERM_CC_DIRECT	GO:0000776∼kinetochore	2	4.21	0.092682	61.726231	
	GOTERM_BP_DIRECT	GO:0000910∼cytokinesis	3	6.31	9.78E−4	1.154575	
	GOTERM_BP_DIRECT	GO:0044772∼mitotic cell cycle phase transition	2	4.21	0.007770	8.844187	
	GOTERM_BP_DIRECT	GO:0051988∼regulation of attachment of spindle microtubules to kinetochore	2	4.21	0.011633	12.969466	
	GOTERM_BP_DIRECT	GO:0031145∼anaphase-promoting complex-dependent catabolic process	2	4.21	0.013559	14.961799	
	GOTERM_BP_DIRECT	GO:0006268∼DNA unwinding involved in DNA replication	2	4.21	0.015482	16.908679	
	GOTERM_BP_DIRECT	GO:0007095∼mitotic G2 DNA damage checkpoint	2	4.21	0.019316	20.670192	
	GOTERM_BP_DIRECT	GO:0007076∼mitotic chromosome condensation	2	4.21	0.021228	22.486821	
	GOTERM_BP_DIRECT	GO:0000070∼mitotic sister chromatid segregation	2	4.21	0.034511	34.093744	
	GOTERM_BP_DIRECT	GO:0001578∼microtubule bundle formation	2	4.21	0.038274	37.079667	
	GOTERM_BP_DIRECT	GO:0000281∼mitotic cytokinesis	2	4.21	0.040151	38.521683	
	GOTERM_BP_DIRECT	GO:0006270∼DNA replication initiation	2	4.21	0.040151	38.521683	
Down-regulated	GOTERM_MF_DIRECT	GO:0005198∼structural molecule activity	9	6.88	5.12E−6	4.557253	
	GOTERM_MF_DIRECT	GO:0004252∼serine-type endopeptidase activity	7	5.35	1.02E−4	0.004512	
	GOTERM_MF_DIRECT	GO:0008201∼heparin binding	6	4.59	1.73E−4	0.187501	
	GOTERM_MF_DIRECT	GO:0005496∼steroid binding	3	2.29	0.003639	3.880030	
	GOTERM_MF_DIRECT	GO:0004962∼endothelin receptor activity	2	1.53	0.015293	11.655613	
	GOTERM_MF_DIRECT	GO:0003707∼steroid hormone receptor activity	3	2.29	0.022939	31.330301	
	GOTERM_CC_DIRECT	GO:0005615∼extracellular space	25	19.12	7.92E−11	7.97E−8	
	GOTERM_CC_DIRECT	GO:0070062∼extracellular exosome	39	29.82	1.45E−10	4.43E−9	
	GOTERM_CC_DIRECT	GO:0005576∼extracellular region	12	9.18	2.40E−6	0.002416	
	GOTERM_CC_DIRECT	GO:0001533∼cornified envelope	5	3.82	4.01E−5	0.040335	
	GOTERM_CC_DIRECT	GO:0005578∼proteinaceous extracellular matrix	7	5.35	2.35E−4	0.236029	
	GOTERM_CC_DIRECT	GO:0045095∼keratin filament	4	3.06	0.002188	2.179044	
	GOTERM_CC_DIRECT	GO:0042567∼insulin-like growth factor ternary complex	2	1.53	0.023333	21.135448	
	GOTERM_CC_DIRECT	GO:0031012∼extracellular matrix	4	3.06	0.023982	21.661062	
	GOTERM_CC_DIRECT	GO:0001527∼microfibril	2	1.53	0.034797	29.965502	
	GOTERM_CC_DIRECT	GO:0042581∼specific granule	2	1.53	0.062878	47.957744	
	GOTERM_CC_DIRECT	GO:0016323∼basolateral plasma membrane	3	2.29	0.087573	60.215342	
	GOTERM_BP_DIRECT	GO:0018149∼peptide cross-linking	5	3.82	6.99E−4	0.098265	
	GOTERM_BP_DIRECT	GO:0030216∼keratinocyte differentiation	5	3.82	3.21E−4	0.451079	
	GOTERM_BP_DIRECT	GO:0007565∼female pregnancy	3	2.29	0.001635	2.274249	
	GOTERM_BP_DIRECT	GO:0008284∼positive regulation of cell proliferation	7	5.35	0.002416	3.344514	
	GOTERM_BP_DIRECT	GO:0045840∼positive regulation of mitotic nuclear division	3	2.29	0.004434	6.057636	
	GOTERM_BP_DIRECT	GO:0048146∼positive regulation of fibroblast proliferation	3	2.29	0.015351	19.550011	
	GOTERM_BP_DIRECT	GO:0006955∼immune response	5	3.82	0.016764	21.157726	
	GOTERM_BP_DIRECT	GO:0001558∼regulation of cell growth	3	2.29	0.018030	22.573586	
	GOTERM_BP_DIRECT	GO:0001755∼neural crest cell migration	3	2.29	0.0189638	23.602523	
	GOTERM_BP_DIRECT	GO:0014826∼vein smooth muscle contraction	2	1.53	0.022138	27.006079	
	GOTERM_BP_DIRECT	GO:0001775∼cell activation	2	1.53	0.033025	37.638886	
	GOTERM_BP_DIRECT	GO:0014068∼positive regulation of phosphatidylinositol 3-kinase signaling	3	2.29	0.034030	38.544070	
	GOTERM_BP_DIRECT	GO:0007267∼cell–cell signaling	3	2.29	0.042835	45.968556	
	GOTERM_BP_DIRECT	GO:0021952∼central nervous system projection neuron axonogenesis	2	1.53	0.043793	46.724165	
	GOTERM_BP_DIRECT	GO:0030198∼extracellular matrix organization	3	2.29	0.048207	50.079668	
	GOTERM_BP_DIRECT	GO:0048484∼enteric nervous system development	2	1.53	0.049132	50.758144	
	GOTERM_BP_DIRECT	GO:0043568∼positive regulation of insulin-like growth factor receptor signaling pathway	2	1.53	0.049132	50.758144	
	GOTERM_BP_DIRECT	GO:0005978∼glycogen biosynthetic process	2	1.53	0.075392	66.786495	
	GOTERM_BP_DIRECT	GO:0006885∼regulation of pH	2	1.53	0.075392	66.786495	
	GOTERM_BP_DIRECT	GO:0010596∼negative regulation of endothelial cell migration	2	1.53	0.075392	66.786595	
	GOTERM_BP_DIRECT	GO:0031290∼retinal ganglion cell axon guidance	2	1.53	0.080557	69.302524	
	GOTERM_BP_DIRECT	GO:0010906∼regulation of glucose metabolic process	2	1.53	0.090803	73.777717	
	GOTERM_BP_DIRECT	GO:0048662∼negative regulation of smooth muscle cell proliferation	2	1.53	0.095883	75.764591	
	GOTERM_BP_DIRECT	GO:0048675∼axon extension	2	1.53	0.095883	75.764591	

Significant pathways identified in CESC

We investigated upregulated and downregulated DEGs to identify the most significantly enriched pathways in each group by GO and KEGG pathway analysis (Fig. 2 and Table 2). With GO analyzing, the results indicated that (1) for biology processes (BP) , the most significantly enriched pathways of the DEGs were epidermis development, positive regulation of cell proliferation, peptide cross-linking, regulation of cell proliferation, positive regulation of cellular process, epidermal cell differentiation, skin development, keratinocyte differentiation, positive regulation of nuclear division, positive regulation of mitotic nuclear division; (2) for molecular function (MF), they were chemokine activity, chemokine receptor binding, calcium ion binding, collagen binding, CXCR chemokine receptor binding, growth factor activity, intergrin binding, cytokine activity, peptidase activity, acting on L-amino acid peptides, CCR chemokine receptor binding; (3) for cell component (CC), DEGs were significantly enriched in spindle, intercalated disc, intermediate filament, mitotic spindle, nuclear chromosome part, spindle midzone, condensed chromosome kinetochore, platelet alpha granule lumen, spindle microtubule and kinesin complex.

The results of KEGG analysis demonstrated that the most significant signaling pathways of DEGs were cell cycle, pathways in cancer, ECM-receptor interaction, arrhythmogenic right ventricular cardiomyopathy (ARVC), melanoma, PI3K-Akt signaling pathway, focal adhesion, vascular smooth muscle contraction, DNA replication and oocyte meiosis (Table 3).

Table 3 KEGG analysis of DEGs in CESC.

Pathway ID	Name	Count	p-Value	Genes	
04110	Cell cycle	13	7.76E−6	PCNA, CDKN2A, BUB1B, CDC7, TTK, SMC1B, CDC20, CCNB1, PTTG1, CDK1, MCM4, MCM5, MCM2	
05200	Pathways in cancer	29	2.77E−5	LAMA2, CKS1B, FGF7, EDNRA, EDNRB, RUNX1T1, PDGFRB, PDGFRA, JUP, CDKN2A, MMP1, ITGA2, PTCH1, FN1, IGF2, MITF, FOS, IGF1, WNT16, GNG11, ESR1, AR, CXCL12, GSTA4, CKS2, BIRC5, FGFR2, GSTM5, FGF10	
04512	ECM-receptor interaction	9	1.36E−4	TNXB, VWF, LAMA2, ITGA2, ITGA8, SPP1, FN1, HMMR, ITGA9	
05412	Arrhythmogenic right ventricular cardiomyopathy (ARVC)	8	2.93E−4	GJA1, LAMA2, JUP, ITGA2, ITGA8, DSG2, DSC2, ITGA9	
05218	Melanoma	8	2.93E−4	PDGFRB, PDGFRA, FGF7, CDKN2A, PDGFD, MITF, IGF1, FGF10	
04151	PI3K-Akt signaling pathway	20	3.18E−4	PDGFRB, PDGFRA, TNXB, VWF, LAMA2, ITGA2, IGF2, FN1, IGF1, GNG11, AREG, EREG, GYS2, FGF7, PDGFD, SPP1, ITGA8, FGFR2, FGF10, ITGA9	
04510	Focal adhesion	13	9.45E−4	PDGFRB, PDGFRA, TNXB, VWF, LAMA2, ITGA2, FN1, IGF1, MYLK, PDGFD, SPP1, ITGA8, ITGA9	
04270	Vascular smooth muscle contraction	10	0.001189	ACTA2, GUCY1A2, PPP1R14A, EDNRA, EDN3, MYH11, MRVI1, AVPR1A, ACTG2, MYLK	
03030	DNA replication	5	0.001491	PCNA, RFC4, MCM4, MCM5, MCM2	
04114	Oocyte meiosis	9	0.002920	CDC20, AR, CCNB1, PTTG1, CDK1, PGR, IGF1, SMC1B,AURKA	

Systematic analysis of core genes by PPI network

PPI network investigated the systematic interaction between the DEGs we got above. Twenty-five DEGs in total were mapped to the DEGs PPI network with 99 nodes and 270 edges. Seven up-regulated DEGs and 18 down-regulated DEGs were included in the PPI network. And then Cytotype MCODE was applied for further analysis of the DEGs in PPI network, and we got a result of 15 particular nodes being identified which were all up-regulated DEGs (Fig. 3).

Figure 3 Common DEGs PPI network constructed by STRING online database and Module analysis.

(A) Nodes meant proteins; the edges meant the interaction of proteins. (B) Module analysis via Cytoscape MCODE tool (degree cutoff = 2, node score cutoff = 0.2, k-core = 2, and max. Depth = 100).

Analysis of core gene signature in CESC using K-M plotter and GEPIA

To investigate the survival data of the 15 genes we identified, K-M plotter indicated that three (TOP2A, RFC4, MCM2) of them had a significant survival rate while other 12 genes had not (P > 0.05) (Fig. 4 and Table 4). The expression of TOP2A, RFC4, MCM2 in normal tissue and CESC tissue was detected by GEPIA. The results showed that the expression of these three genes in CESC tissue was higher than that in normal tissue (P < 0.05) (Fig. 5).

Figure 4 The prognostic information of the 15 core genes.

Three (A, B and C) of 15 genes had a significant better survival rate (P < 0.05) and twelve genes (D–O) had not significant difference in OS (P > 0.05).

RFC4 is validated to be overexpressed in CESC

By analyzing the data from the NCBI-GEO dataspace for mRNA expression in CESC patients, RFC4 gene was identified as an overexpressed gene in CESC patients. We collected 35 pairs of CESC patients for qPCR, the tissues of 6 pairs CESC patients were used for WB, 9 pairs CESC tissues and 4 normal cervical tissues for IHC. In order to validate our finding, total RNA was extracted from 35 paired CESC tissues and para-CT tissues, and qRT-PCR was conducted to measure the expression level of RFC4 gene. The result showed that the expression level of RFC4 on CESC tissues was significantly high compared with para-CT (P = 0.0197) (Fig. 6). And the expression of RFC4 in early stage CESC was significantly higher than that in advanced CESC (P = 0.0314) (Fig. 7). The same result was invested from WB. The results of WB analysis indicated that the RFC4 was overexpressed in CESC tissues compared to para-CT tissues (Fig. 8). A higher level of RFC4 expression on CESC tissues was observed from the result of IHC, and RFC4 protein was mainly concentrated in the nucleus (Fig. 9).

Table 4 The information of prognostic analysis of 15 core DEGs.

Category	Genes	
Genes with significant (better) survival (P < 0.05)	TOP2A RFC4 MCM2	
Genes without significant survival (P > 0.05)	UBE2C PRC1 NUSAP1 NEK2 MCM5 KIF20A HMMR FANCI ECT2 DTL AURKA ASPM	

Figure 5 Expression level of three significantly expressed genes in CESC tissues and normal tissues.

(A) The expression level of TOP2A in CESC tissues and normal tissues. (B) The expression level of MCM2 in CESC tissues and normal tissues. (C) The expression level of RFC4 in CESC tissues and normal tissues. Red color means tumor tissues and grey means normal tissues.

Figure 6 The expression of RFC4 on CESC was significantly different compared with para-cancerous tissues from the result of qRT-PCR.

Figure 7 Expression levels of RFC4 in different pathological stages of CESC.

The expression of RFC4 in early stage CESC was significantly higher than that in advanced CESC (P = 0.0314).

Figure 8 WB analysis of RFC4 protein.

C: CESC tissues, P: para-cancerous tissues. (A) Six pairs CESC tissues WB analysis indicated that except that the results of case 4 are not obvious, the others are consistent with the expected results of high expression of RFC4 in tumor tissues. (B) The grayscale analysis of multiple WB bands shows that the WB tests are reliable.

Figure 9 IHC test of CESC.

IHC declared that, in general, the RFC4 protein is highly expressed in tumor tissue sections, and is mainly concentrated in the nucleus, while normal cervical tissue and para-cancerous tissues are underexpressed.

Discussion

In order to identify more effective prognostic biomarkers in CESC, we used different bioinformatics methods to analyze three data sets based on NCBI-GEO database, including 92 CESC tissues and 36 normal tissues. A total of 25 DEGs were selected by GEO2R and Venn software, including seven up-regulated genes and 18 down-regulated genes. Then GO and KEGG pathway analysis were conducted, and the results of GO and KEGG indicated that the selected DEGs were significantly enriched in various cell pathways. Research reported that genes from these pathways could be associated with the pathogenesis and progression of cervical cancer. Nucleolar and spindle associated protein 1(NUSAP1) was a gene from spindle associated pathway, and it was reported to promote the metastasis of cervical cancer by activating Wnt/β-catenin signaling (Li et al., 2019). And studies showed that CXCL12/CXCR4 pathways was associated with HPV infection as a co-factor, which means a high risk to the incidence of cervical cancer (Meuris et al., 2016). Genes involved epidermis development were also associated with the high-risk HPV infection (Zhang et al., 2018; Chatterjee et al., 2019).

After that PPI network was constructed using STRING software and MCODE analysis was conducted, and 15 particular DEGs were identified. Furthermore, by K-M plotter analysis we found three DEGs from the 15 which had a significantly better survival. The results of GEPIA showed that the expression levels of the three selected genes in CESC tissues were higher than that in normal tissues. To further validation, we performed RFC4 relevant molecule biological experiments and the results showed that compared with normal tissues, RFC4 was highly expressed in CESC tissues.

Being short for Replicant Factor C, RFC is a structure specific DNA- binding protein acting as a primer recognition factor for DNA polymerase (Zhou & Hingorani, 2012), which includes five subunits (RFC1-5). Among all five subunits of RFC complex, RFC4 has been reported to play an important role in DNA damage checkpoint and DNA replication pathways (Ellison & Stillman, 2003). In 2009, Arai M et al. reported that RFC4 was closely related to the prognosis of liver cancer (Arai et al., 2009). Besides liver cancer, RFC4 has been reported to be associated with several types of cancer, including prostate cancer, colon cancer non-small cell lung cancer and leukemia (LaTulippe et al., 2002; Jung, Choi & Kim, 2009; Erdogan et al., 2009; Barfeld et al., 2014). Research illustrated that up-regulated RFC4 expression found in neck squamous cell carcinoma and it was 3.4-fold higher than that in normal tissues (Slebos et al., 2006). Studies from Garnett et al. (2012) showed that RFC4 can be regulated by mutated RB1 in several types of cancers, suggesting that RFC4 could be a potential biomarker associated with the occurrence and prognosis of various cancers. Moreover, RFC4 was reported as an independent predictor of overall survival in breast cancer (Fatima et al., 2017; Niu et al., 2017).

In this study we observed RFC4 as a potential independent prognostic biomarker in CESC, and our results suggested that CESC patients with higher expression level of RFC4 may have a better overall survival. A possible reason might be that RFC4 was highly expressed throughout the cell circle process of proliferating cells, and tumor proliferation in situ will become slow with the development of the disease (Szymanska et al., 2018; Chaplain & Sleeman, 1993), which means a decrease in the expression of RFC4. Therefore, highly expressed RFC4 may suggest early stage CESC, which indicates better overall survival.

Several studies have proved that these three genes were associated with numerous types of cancer, but studies of RFC4 in CESC were rarely seen, and very few researches conducted molecule biology validation. Therefore, our study shows that RFC4 is a potential biomarker for the predicting the prognosis of CESC and provides a direction for further study of CESC. What should be noted is that there are some limitations in this study. Clinical samples from one hospital may have either region or race difference. The expression level of RFC4 in different stages of CESC and clinical investigations should be conducted in our future study to validate our results further.

Conclusions

In conclusion, by using bioinformatics analysis we identified three genes (TOP2A, RFC4, MCM2) based on three microarray datasets. These three genes were suggested to have a significant effect on the prognosis of CESC, which could be key factors in the occurrence and progression of CESC. A high level expressed RFC4 was validated to exist in CESC tissues using clinical samples. Although further investigation and experiments needs to be conducted, the findings in our study could act as clinical biomarkers which would help us better understand the pathological process and predict the prognostic of CESC.

Supplemental Information

Supplemental Information 1 Raw data for PCR

Click here for additional data file.

Supplemental Information 2 Full Length Uncropped Blots

Click here for additional data file.

Supplemental Information 3 Raw data for western blot greyscale analysis

Click here for additional data file.

Supplemental Information 4 Raw data for PCR of different pathological stages

Click here for additional data file.

The authors appreciate the patients who provided tumor tissue for this study. We also thank to Xiaofan Li for English editing.

Additional Information and Declarations

Competing Interests

Author Contributions

Human Ethics

Data Availability

The authors declare there are no competing interests.

Yunan He conceived and designed the experiments, analyzed the data, prepared figures and/or tables, authored or reviewed drafts of the paper, and approved the final draft.

Shunjie Hu analyzed the data, prepared figures and/or tables, authored or reviewed drafts of the paper, and approved the final draft.

Jiaojiao Zhong performed the experiments, analyzed the data, prepared figures and/or tables, authored or reviewed drafts of the paper, and approved the final draft.

Anran Cheng conceived and designed the experiments, authored or reviewed drafts of the paper, and approved the final draft.

Nianchun Shan conceived and designed the experiments, performed the experiments, prepared figures and/or tables, and approved the final draft.

The following information was supplied relating to ethical approvals (i.e., approving body and any reference numbers):

The Medical Ethics Committee of Xiyang Hospital approved this research (No.201912542).

The following information was supplied regarding data availability:

The main data of this study are available at NCBI-GEO: GSE27678, GSE39001, GSE7803. The raw data are available as Supplemental Files.

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
