# Peer review of "Identification of significant genes signatures and prognostic biomarkers in cervical squamous carcinoma via bioinformatic data"

_PeerJ, doi:10.7717/peerj.10386_

## Round 0.1 · original submission · Major Revisions

Dear Dr. He,

The comments from the reviewers are just received. Please make some significant updates as the reviewers' suggested, and you are encouraged to re-submit your updated manuscript.

Thanks!
Jianye Ge

Reviewer 1 ·

Basic reporting

(1) Figures and tables need more improvement. Figure 1 shows wrong aspect ratio. The texts of last few lines in Figure 2 are hard to tell between the background color. Almost all figures have low resolution. Especially, the label of every node in Figure 3A isn’t clearly. Column Table 1 needs better description. E.g. “Expression” and “Genes Name”.

(2) Section “Materials & Methods” shows lots of bioinformatics analysis methods but lacks the citation of them.

Experimental design

Major comments:
(1) There is a lack of innovation in integrated bioinformatical analysis part. The main analysis pipeline is highly similar with Identification of key candidate genes and small molecule drugs in cervical cancer by bioinformatics strategy (https://doi.org/10.2147/CMAR.S171661).

(2) In section “Identification of differentially expressed genes”, you choose DEGseq as the DEG analysis tool. First, the citation of DEGseq is incorrect. You must check all citations carefully! Then, As “How many biological replicates are needed in an RNA-seq experiment and which differential expression tool should you use?” (https://doi.org/10.1261/rna.053959.115) said, DEGseq has unsatisfied effect between these similar tools such as edgeR and DESeq2. So could you explain why you choose DEGseq in this section?

Minor issues


(1) It needs more solid results to support the discuss. For example, the conjecture “highly expressed RFC4 may suggest early stage CESC, which indicates better overall survival” needs further comparison about the expression of RFC4 in samples at different stages.

Validity of the findings

no comment

Additional comments

This paper identified three differentially expressed genes on the prognosis of CESC by integrated bioinformatical analysis such as Gene Ontology, KEGG pathway analysis, PPI network and Kaplan-Meier survival analysis. Then, RFC4 was validated to be significantly associated with the occurrence and prognosis of CESC using clinical samples by quantitative transcription PCR, Western Blot and immunohistochemistry. Although the clinical validation of RFC4 has scientific merit, this paper needs further improvement to be ready for publication.

Annotated reviews are not available for download in order to protect the identity of reviewers who chose to remain anonymous.

Reviewer 2 ·

Basic reporting

In this manuscript, the authors focused their attention in the identification of "significant gene signatures and prognostic biomarkers in CESC via integrated bioinformatical analysis". Although the authors used a reliable database (GEO) and molecular assays were conducted to validate their findings, I do have some questions and comments for the authors. In particular, extensions, modifications and better explanations of their methods and results are suggested. In addition, a linguistic revision is needed.

Experimental design

1. The manuscript is entitled "Identification of significant genes signatures and prognostic biomarkers in cervical squamous carcinoma via integrated bioinformatical analysis". What does "signficant genes signatures" means? Is it means "differentially expressed genes"? Furthermore, the microarray data obtained from GEO seems not original data, and the authors do not need to process the data by any bioinfomatics methods. Therefore, it should be a data mining analysis. For the above reasons, the title should be modified.
2. For K-M survival analysis, the authors should illustrate how to discretize the continuous gene expression. Furthermore, to estimate the hazard ratio of gene expression on patient survival, the authors should check the proportional hazards assumption.
3. The follow-up data should be provided, so that the readers can replicate the findings of the manuscript.
4. For the differential expression analysis, the standard should be consistent and do not need to described twice (in the methods and results).
5. The results of the enrichment analysis should be well explained, not merely list them out.

Validity of the findings

This work find that 3 genes were differentially expressed and associated with patient survival. Furthermore, the authors validated 1 gene in clinical specimens form RNA and protein level. Overall, this study is lack of innovation and workload.

Additional comments

no comment

---

## Round 0.2 · accepted · Accept

Congratulations! Based on the reviewer's comments, we would like to accept your revised manuscript for publication.

Reviewer 1 ·

Basic reporting

no comment

Experimental design

no comment

Validity of the findings

no comment

Additional comments

It can be accepted.